# Molecular Distillation of Lavender Supercritical Extracts: Physicochemical and Antimicrobial Characterization of Feedstocks and Assessment of Distillates Enriched with Oxygenated Fragrance Components

**DOI:** 10.3390/molecules27051470

**Published:** 2022-02-22

**Authors:** Agnieszka Dębczak, Katarzyna Tyśkiewicz, Zygmunt Fekner, Piotr Kamiński, Grzegorz Florkowski, Marcin Konkol, Edward Rój, Agnieszka Grzegorczyk, Anna Malm

**Affiliations:** 1Łukasiewicz Research Network—New Chemical Syntheses Institute, Al. Tysiąclecia Państwa Polskiego 13A, 24-110 Puławy, Poland; katarzyna.tyskiewicz@ins.lukasiewicz.gov.pl (K.T.); zygmunt.fekner@ins.lukasiewicz.gov.pl (Z.F.); piotr.kaminski@ins.lukasiewicz.gov.pl (P.K.); grzegorz.florkowski@ins.lukasiewicz.gov.pl (G.F.); marcin.konkol@ins.lukasiewicz.gov.pl (M.K.); edward.roj@ins.lukasiewicz.gov.pl (E.R.); 2Chair and Department of Pharmaceutical Microbiology, Medical University of Lublin, 1 Chodźki Str., 20-093 Lublin, Poland; agnieszka.grzegorczyk@umlub.pl (A.G.); anna.malm@umlub.pl (A.M.)

**Keywords:** antibacterial and antifungal activity, coumarins, lavender, molecular distillation, natural wax ingredients, oxygenated terpenes, pilot-scale SFE

## Abstract

*Lavandula angustifolia* is one of the most widely cultivated non-food crops used in the production of essential oil; it is used in perfumery, aromatherapy, pharmaceutical preparations, and food ingredients. In this study, supercritical fluid extraction (SFE) and molecular distillation (MD) were combined, primarily to enrich scCO_2_ extracts with lavender oxygenated monoterpenes, avoiding thermal degradation, hydrolysis, and solvent contamination, and maintaining the natural characteristics of the obtained oils. Molecular distillation was developed for the first time for the extraction of crucial lavender fragrance ingredients, i.e., from two scCO_2_ extracts obtained from dry flower stems of lavender cultivated in Poland and Bulgaria. The best results for high-quality distillates were obtained at 85 °C (EVT) and confirmed that linalyl acetate content increased from 51.54 mg/g (initial Bulgarian lavender extract, L-Bg-E) and 89.53 mg/g (initial Polish lavender extract, L-Pl-E) to 118.41 and 185.42 mg/g, respectively, corresponding to increases of 2.3 and 2.1 times in both distillate streams, respectively. The distillates, light oils, and extracts from lavender were also evaluated for their antimicrobial properties by determining the minimum inhibitory concentration (MIC) by the broth microdilution method. Generally, Gram-positive bacteria and *Candida* spp. were more sensitive to all distilled fractions and extracts than *Escherichia coli* (Gram-negative bacteria).

## 1. Introduction

Lavender (*Lavandula angustifolia* Mill.), recognizable as medicinal lavender, is a known source of precious fragrance molecules, with predominating linalyl acetate and linalool, which belong to the group of oxygenated monoterpenes, representing 73% of essential oil compositions [1]. Both compounds are responsible for a wide range of pharmacological effects, such as anti-inflammatory, antifungal, antibacterial, anti-viral, antioxidant, cicatrizant properties, and anxiety-reducing [2,3,4,5,6]. Moreover, the stress-relieving and sleep-promoting benefits of lavender oil are therapeutically used in aromatherapy and massages [5,7]. The benefits of essential oils from the *Lavandula* include their strong antiseptic properties, which make this specialty oil applicable, directly to pharmaceuticals and cosmetics, most of which are related to skincare products [8]. The latest studies on the potential of *L. angustifolia* essential oil revealed its beneficial impact on melanogenesis and its inhibition of tyrosinase, which can be attributed to one of the lavender oxygenated monoterpenes, namely terpinen-4-ol [9].

Due to a high demand for linalyl and lavandulyl acetates, which, together with linalool and lavandulol, are of special interest for cosmetic and perfumery applications, the attention is placed on the effective purification technology of these valuable compounds. The conventional/fundamental—and often used extraction methods of lavender essential oils—involve steam distillation and hydrodistillation as its variants. However, there are several crucial aspects regarding these two extraction methods (steam and hydrodistillation), including a proper time to release valuable compounds without the influence on their structures [10]. The effect of prolonged steam contact is reflected in product composition, especially caused by the hydrolysis of linalyl acetate to linalool [11]. The use of modern extraction methods, for instance, with carbon dioxide in a supercritical state (SFE), offers the extraction of biologically active compounds without a negative impact on their properties and, more importantly, on the environment [12]. Since the most precious components of *Lavandula angustifolia* Mill. are oxygenated compounds, including derivatives of monoterpenes, i.e., esters and alcohols (linalool, lavandulol, linalyl acetate, lavandulyl acetate), sesquiterpenes, and lactones (coumarins), SFE has been utilized for the extraction of *Lavandula angustifolia* of different origins (Bulgarian and Polish). According to Jerkovic et al. [13], the highest total extraction yield (7.28%) and highest relative amount of oxygenated monoterpenes, i.e., linalool and linalyl acetate from *L. angustifolia,* was obtained at 30 MPa and 50 °C. Nadalin et al. [14] highlighted that by increasing the scCO_2_ pressure from 10 to 30 Mpa, with a constant temperature (40 °C), the extraction yield also increased (from 5.16 to 7.08 g/100 g D.W. (dry weight)). Moreover, extraction is the first step toward separating bioactive compounds and, thus, may be considered as a pre-stage for further separation [15].

The vacuum distillation and SFE fractionations were first applied for the production of high quality oxygenated compounds (linalool and linalyl acetate) from cold-pressed bergamot oil, using the former technique mostly for the removal of limonene [16]. The relatively low recovery of oxygenated compounds was obtained with non-continuous batch distillations on an industrial scale, as waxes, pigments, and oxygenated fragrance molecules were not efficiently separated in the system due to high viscosity and, thus, remained in residual vessel [16]. The higher effectiveness may be achieved with the use of the molecular distillation than with conventional methods [17]. The combination of a very short residence time (1–10 s) of a separated product within a distillation unit, a continuous mass transfer of evaporated molecules traveling in a free flow regime through the distillation gap (10–50 mm) onto the surface of the condenser, and high vacuum levels in the 0.1333–133.3 Pa range [18,19], were successfully applied for the separation and purification of thermally unstable products preserving their functional properties. The influence of molecular distillation operating parameters, primarily the evaporator temperature (EVT) and feed flow rate (FF), was thoroughly studied in order to increase the concentration of precious scent ingredients, i.e., citral [20], geranial [21], linalool [22], and sesquiterpenoids, i.e., patchouli alcohol [23], which all belong to oxygenated compounds. The major ingredient of the citrus oil light terpenic fraction recovered in the distillate was D-limonene [21,22]. Hydro-distilled essential oils derived from rosemary [24], oregano [25], and rose [26], mostly known for their antimicrobial and antioxidant activities, were subjected to molecular distillation. The obtained fractions showed higher antioxidant and antimicrobial activities relative to their original essential oils; hence, they may be used as natural additives for food preservation [25] or plant-based pharmaceuticals and healthcare products [26]. Some challenging applications of molecular distillation involve further purification and fractionation of supercritical fluid extracts to prepare standardized typical volatile components fractions from *Curcuma* species [15] or separate essential oils and wax-like residues from *Artemisia annua* L. scCO_2_ extracts, depending on the optimized extraction parameters [27]. Pilot scale supercritical CO_2_ extraction coupled with molecular distillation (SFE-MD) was recently applied for the separation of essential oils from *Artemisia argyi* Lévl. Et Vant [28]. Currently, molecular distillation is considered advantageous, in regard to purifying high molecular weight compounds without thermally degrading the products, such as rice bran wax unsaponifiable components [29]. One of the newest demanding applications of MD involves the fractionation of polyethylene wax, which, performed on a pilot scale, is reported to be a promising solution for recycling and recovery of high value-added products with narrower carbon numbers [30].

In this study, the supercritical fluid extraction and molecular distillation were combined primarily to enrich scCO_2_ extracts with lavender oxygenated monoterpenes, avoiding thermal degradation or hydrolysis and solvent contamination, and maintaining the natural characteristics of the obtained oils. Molecular distillation was developed for the first time for the extraction of crucial lavender fragrance ingredients, i.e., from two scCO_2_ extracts obtained from dry flower stems of *Lavandula angustifolia* Mill. cultivated in two European countries: Poland and Bulgaria. The scCO_2_ extracts (L-Pl-E, L-Bg-E) were thoroughly studied according to their physicochemical and antimicrobial properties. In order to evaluate the influence of evaporator temperature (from 55 to 95 °C) under vacuum (10^−2^ mbar) and a constant feed flow rate (0.833 mL/min) on the recovery of the target components, gas chromatography equipped with flame ionization detection (GC–FID) analysis was applied. In this study, the in vitro antimicrobial (antibacterial and antifungal) activity of distillates, light oils, and extracts from *L. angustifolia* Mill. were also investigated.

## 2. Results

### 2.1. Composition of Lavender scCO_2_ Extracts

Over the last decade, the quality of lavender scent extracted by scCO_2_ in the pressure ranges of 10–30 MPa and up to 40 °C were reported on in literature, utilizing either a pilot scale [11,13] or laboratory scale SFE units [13,14]. The supercritical extraction of lavender was studied by Nadalin et al. [14]. It was shown that maintaining the pressure at 30 MPa and increasing the temperature to 59 °C resulted in the highest yield of lavender extract (around 7.5 wt%) [14]. In accordance with this reported data, a similar extraction yield was obtained for the extraction of *Lavandula angustifolia* of Polish origin (L-Pl) in this paper, resulting in 7.05 wt% as a ratio of the obtained extract mass to the feed mass for the extraction. However, the extraction temperature was significantly lower (40 °C) compared to that applied by Nadalin et al. [14]. The Bulgarian *L. angustifolia* (L-Bg) was characterized by slightly lower extraction yield (6.33 wt%) under analogous extraction conditions. The laboratory-scale investigations mostly aimed to optimize the lavender extraction process, in terms of a high total extraction yield and a relatively high amount of linalool, lavandulol, linalool acetate, and lavandulol acetate [13,14]. The increase of lavender flower extraction yield at a constant temperature (40 °C) was noticed when increasing the pressure from 10 to 30 MPa [14]. Under these conditions, higher molecular weight components, such as oxygen-bearing molecules, e.g., coumarins [13], polyphenols [31], and non-volatile cuticular waxes [32], may be co-extracted together with fragrance volatiles [32,33].

The compositions of lavender supercritical fluid extracts have been thoroughly studied in relation to extraction parameters and their impacts on the yields of advantageous flavor components [13,14,34]. With the pressure increasing up to 30 Mpa at 40 °C, the CO_2_ density also increases, which favors higher extraction yields and the apparent solubility of the most characteristic group of lavender odoriferous molecules—oxygenated monoterpenes, i.e., ester or alcohol components [14]. Unavoidably, targeted fragrance molecules are co-extracted with waxy, resinous, and color matter; hence, the product is concrete-like (Figure 1). The scCO_2_ lavender extract is enriched with scent volatile compounds much more than the oil [35] and, due to preservation of initial plant matter, it can be served as a valuable feedstock for further fractionation and valorization processes [13].

The extract compositions from the L-Bg-E and L-Pl-E samples were determined by means of the GC–MS qualitative analysis. Results of GC–MS identification and relative amounts of components (%) are listed in Table 1. Additionally, chromatograms corresponding to each extract are depicted in Figure 2.

The GC–MS analysis revealed the presence of more than sixty different compounds in the extracts. (Table 1, Figure 2). Linalool and linalool acetate, the main representatives of lavender oxygenated monoterpenes, were the most abundant components; however, their percentages differed notably from one extract to another. The targeted compounds represented the lowest percentage composition in the case of the L-Bg-E sample with 4.56% and 12.78% for linalool and linalool acetate, respectively (Table 1). The highest percentages of linalool; linalool acetate and lavandulol; lavandulol acetate and terpinen-4-ol were confirmed in the cases of L-Pl-E (17.02; 25.68; 1.40, 3.96 and 6.02%, respectively) (Table 1). Lavandulol and its acetate are of a great interest in the cosmetic and perfumery industries, as it gives the oil a rosaceous, sharp floral aroma [36,37]; thus, their higher levels in natural oil can strongly increase its price. Together with linalool, linalyl acetate, terpinen-4-ol, and α-terpineol—the compounds were recently selected to study the influence of long-term storage on essential oil content and in the quality of two Czech *Lavandula angustifolia* Mill. lavender varieties [38]. The gradual lowering of the total content of essential oils (2.56% per year) was noticed; however, no statistically significant relationship between their initial compositions and the loss of the selected fragrance key markers in longer-term monitoring were found.

Linalool, one of the most important compounds for the perfume and flavor industries, occurs naturally in *Lavandula angustifolia* Mill., as two isomers, almost exclusively predominating ^®^(-)-linalool (94.1%) [39]. This enantiomer is a flowery-fresh woody odor, reminiscent of lavender, which, together with an array of its derivatives, is a common constituent of flower and honey extracts obtained from different lavender and citrus sources [40]. Both L-Bg-E and L-Pl-E are characterized by the presence of 8-hydroxylinalool (2,6-dimethyl-2,7-octadiene-1,6-diol) resulting directly from the activity of P450 hydroxylase. Two isomers of 8-hydroxylinalool (*E*-; *Z*-) were found to be direct hydroxylation products of linalool, and together with α-terpineol, limonene, and nerolidol, were included in citrus honey, indicating their direct delivery from flower nectar [41]. Other products of direct linalool hydroxylation were 8-oxolinalool and 6,7-epoxylinalool [42]. In the lavender extracts studied, *cis*- and *trans*-linalool oxides (furanoid) and terpendiol isomers, i.e., 3,7-dimethyl-1,7-octadien-3,6-diol, were analyzed as products originating from further (consecutive) enzymatic reactions of 6,7-epoxylinalool (Table 1, Figure 3).

These linalool bioconversion products were previously detected in supercritical lavender extracts obtained under the pressure of 30 MPa and temperatures of 40–50 °C [13]. Another component previously specified in plant and honey extracts of leatherwood (*Eucryphia lucida* (Labill.)) and *Citrus* spp. is 2,6-dimethyl-3,7-octadiene-2,6-diol (precursor of hotrienol) [40], also determined in L-Bg-E and L-Pl-E at the level of 2.81 and 1.34%, respectively.

Eucalyptol (1,8-cineole) represents the oxygenated monoterpenes providing medicinal and olfactory properties of fine lavender essential oils (*Lavandulae aetheroleum*) [10]. The richest lavender oils, in terms of the content of such compounds as 1,8-cineole and camphor, are spike lavender oil and lavandin oil Grosso var. [43]. The camphor-based products may be utilized for industrial cleaning products, detergents, or even natural and healthier alternatives to traditional solvents commonly used in fine art painting. However, in *Lavandulae aetheroleum,* camphor content should not exceed 0.5% according to EU PDO specifications [43]. High concentrations of camphor, as well as 1,8-cineole and terpinen-4-ol, can adversely affect the quality of lavender natural oils. However, the antimicrobial and antifungal bioactivities for linalool, 1,8-cineole, camphor, terpineol, and isomers of pinene, was proven [44,45].

Other valuable substances of industrial interest, which were found in scCO_2_ lavender extracts, are sesquiterpenes, hydrocarbons, and their oxygenated forms, including alcohols. The components, i.e., β-caryophyllene, β-caryophyllene oxide, α-santalene, α-santalol, β-santalol, β-farnesene, α-bergamotene, nerolidol, α-bisabolol, tau-cadinol, ledene oxide-(II), were analyzed in both studied extracts (Table 1). However, interpretation in the sesquiterpene region of the chromatogram may be imprecise due to structural similarities of eluted components. β-Caryophyllene oxide is a predominant sesquiterpenoid representative found in the studied extracts with percentages of 3.80% (L-Pl-E) and 4.54% (L-Bg-E). Both β-caryophyllene and its oxidation product β-caryophyllene oxide are used as cosmetics and food additives since they are approved as fragrance supplements by regulatory authorities, such as the FDA and EFSA [46]. It was reported that SFE conditions enable high concentrations of caryophyllene oxide, linalool, and geranial at 60 °C and under pressure of 30 MPa [47].

The studied lavender extracts contained other groups of oxygen-bearing components, i.e., lower aliphatic compounds: 1-octen-3-ol, octan-3-one, and characteristic lactones. The last group of lavender compounds includes coumarins and lavender lactone (5-ethenyldihydro-5-methyl-2(3H)-furanone), which is known for its interesting olfactory properties [13]. Along with coumarin, its derivatives (i.e., herniarin) are recognized from their significant pharmacological properties, e.g., antitumor, anti-inflammatory, antibacterial, antifungal, diuretic, analgesic, and cardiovascular [48]. Furthermore, natural coumarin is a desirable ingredient due to its sweet, distinguishing vanilla-like scent with grassy notes, applied either in masculine or feminine perfume compositions, and recognizable as a tonka bean fragrance [44]. *Lavandula angustifolia* is another source of coumarin and herniarin, extracted from flowers, with the use of supercritical CO_2_. The quantitative analysis of coumarin and its 7-methoxy derivative in L-Bg-E and L-Pl-E samples, as well as distilled fractions, was performed by supercritical fluid chromatography (SFC). The less soluble and non-volatile paraffins included in the cuticle layer of flower stems are easily co-extracted (“superficial washing”); hence, finally, a higher boiling fraction dilutes the fragrance molecules in this complex matrix [32,33]. According to the GC–MS qualitative analysis, a similarity between the chemical composition of the paraffin fraction found in L-Bg-E and L-Pl-E was found, with predominating hentriacontane (C31) and tritriacontane (C33) in both extracts (Table 1).

### 2.2. Thermal Properties of scCO_2_ Extracts

Thermal characteristics of lavender extracts were determined by differential scanning calorimetry (DSC). The DSC thermograms recorded during the second heating of L-Bg-E and L-Pl-E samples are depicted in Figure 4 and compared with melting profiles of commercial waxes: carnauba and beewax. Supercritical extracts of *L. angustifolia* Mill. from two European cultivars, Polish and Bulgarian, have been scarcely studied and compared according to the compositions of their waxy fractions.

The DSC profiles of L-Bg-E and L-Pl-E showed two endotherms in each analyzed sample (Figure 4). The melting points of L-Bg-E were at 58 °C and 81 °C, while that of L-Pl-E appeared to be at 52 °C as the most intensive peak and 75 °C at a much lower intensity. Both thermograms may suggest coexistence of two different waxy structures in the lipid mixtures of lavender scCO_2_ extracts and, thus, their multicomponent natures. Lower melting fractions signified by the first peaks (58 °C and 52 °C) in the heat flow curves of L-Bg-E and L-Pl-E, respectively, were also found at DSC curves of beewax (55 °C) and carnauba wax (58 °C), (Figure 4). In the case of a beewax heat flow curve, the peaks at 40–60 °C were attributed to the heat absorption of free fatty acids (13%) and hydrocarbons (13%) [49]. Both groups of components were qualified in lavender scCO_2_ extracts with predominating oleic and linoleic fatty acids (analyzed as methyl esters), and long-chain alkanes (C31–C33), Table 1.

The DSC scans of both lavender extracts did not give endotherms at 60 °C to 70 °C, which could approve fatty acids esters. This fraction was abundant in beewax (72%) and led to the sharp absorbing peak at 60–70 °C [49].

Carnauba wax is noticeably different than beeswax and its chemical composition and physical properties are discussed elsewhere [50,51,52]. Carnauba wax contains major proportions of esters of hydroxy acids, i.e., *p*-methoxycinnamic diesters (PCO-C), which isolates as a pure fraction, showing up in the DSC curve as an endothermic peak at 73 °C with a shoulder at 66 °C [53]. Similar structures could be responsible for some minor features in the second heat thermal analysis of L-Pl-E found at 75 °C (Figure 3). However, further increase of melting properties was supposed to be affected by the addition of methoxy substituent on the phenol ring, and such behavior was observed for *p*-coumaric acid esters, i.e., methyl ferulate and methyl sinapate, showing fusion points of 62 and 88 °C, respectively [54]. This might support the similarity between the highest melting fractions of L-Bg-E (81 °C) and carnauba wax (83 °C) in terms of carboxylic acid constituents (Figure 4).

The difference between L-Bg-E and L-Pl-E reflected in DSC thermograms at the most intensive melting points may indicate that the waxy structures of both *L. angustifolia* raw materials depend on its geographic source. Since Bulgarian lavender cultivars are grown on the sunny slopes in a warm and dry climate, the compositions of their cutin layers, covering flowering stems and petals of flowers, give the film excellent barrier properties against UV radiation and uncontrolled water loss. The presence of a higher melting fraction depicted as a sharp peak at 81 °C (Figure 4) may suggest an increased proportion of phenol constituents lipophilized through its esterification. The isolation of those bioavailable natural components with antiradical properties can widen diverse uses of lavender, acknowledged in the cosmetic and pharmaceutical industries.

### 2.3. Application of Molecular Distillation in Lavender scCO_2_ Extract Fractionation

Molecular distillation is a type of vacuum distillation that separates components of a mixture by their difference in volatility. A distillation unit designed for short-path operations (molecular distillation) is made up of evaporator and internal spiral condensers placed in its center. The distance from the evaporation surface to the condensation surface does not exceed the mean free path of molecules, which means that, once evaporated, they reach the condensing surface without delay. Additionally, due to specific working conditions provided in the system by a vacuum pump set (0.1–100 Pa), the relative volatility of compounds increases, which allows separation of mixture components at lower temperature. For the optimal product distribution, the feed product is pumped on top of a rotating wiper basket plate, before it is mixed to a thin film by a wiper system. The most important for the system is that it ensures a short residence time of complex mixtures in the distillation unit and largely prevents thermal decomposition of their components, divided during the process into distillate and/or residue streams.

The objective of the present study was to evaluate the potential of a selected approach, to incorporate fractionation under vacuum into the refinement of scCO_2_ produced lavender extracts. Molecular distillation as a gentle technique of physical refining was applied to separate essential oil components from heavy components co-extracted with scCO_2_ on a pilot scale. Hence, it was feasible to recover higher yields of precious oxygenated monoterpenes from scCO_2_ extracts enriched under the selected conditions (30 MPa, 40 °C). The SFE-MD strategy was already found to be more efficient in a separation of essential oils from artemisia argyi Lévl. Et Vant, compared to hydrodistillation described as time- and energy-consuming and destructive for thermolabile compounds (high water abundance and high process temperatures leading to hydrolysis) [28]. What is more important, the latter mentioned disadvantage is crucial to overcome especially in relation to recovery of linalyl and lavandulyl acetates. Caryophyllene oxide is one of the highest boiling constituents identified in the lavender essential oil analyzed with GC–MS [55]. However, the boiling point of lavender essential oil (204 °C) as a mixture is a function of the vapor pressures of its major and minor volatile constituents. Coumarin and herniarin are representatives of lactones, which, similar to monoterpenoids, are abundant components of scCO_2_ extracts obtained under increased extractant density [13]. Thus, the highest boiling components from the studied group of volatiles can also feed the distillate stream, depending on the applied molecular distillation conditions.

The essential oils included in the scCO_2_ lavender extracts (Figure 1) are diluted with higher boiling components with a waxy character, causing congelation at room temperature. The studied feedstocks differed slightly in densities between L-Bg-E and L-Pl-E (0.98 g/mL and 0.93 g/mL, respectively) and in composition/proportion to a higher melting fraction, as confirmed by the DSC curve of L-Bg-E (81 °C). Those features of lavender extracts may further impact the performance of thin film evaporator. The boiling points of the studied volatiles are listed in Table 2.

For the purpose of the experiment, fractionation was performed under the pressure down to 1 Pa and the temperature below 100 °C, to assure mild processing conditions for major constituents of lavender aroma. The rest of the crucial parameters were checked and adjusted individually to each extract. The experimental conditions of molecular distillation processes performed on L-Bg-E and L-Pl-E feedstocks are listed in Table 3.

Accordingly, as can be seen in Table 3, the temperatures of the feed tank (FT) were different in order to keep L-Bg-E and L-Pl-E in a liquid form at 50 °C and 45 °C, respectively. The values of the evaporator temperatures (EVT) and condenser temperature (CTs) were chosen individually to each of the studied lavender scCO_2_ extracts, based on the first distillate drops falling down the condenser. Since there was no oil distilled out at 50 °C and below that temperature, 55 °C was chosen as the initial EVT for both extracts, and five experiments with evaporator temperatures ranging from 55 °C to 95 °C, with an interval of 10 °C, were performed with a constant feed flow (FF) rate of 0.833 mL/min on L-Bg-E and L-Pl-E (Table 3). The CT temperature was kept constant at 10 °C during the entire distillation experiment of the L-Pl-E extract, while it was varied from −5 °C to 6 °C in distillations D1–D5 in the case of L-Bg-E fractionation. Additionally, the residue discharge temperature (RdT) was raised in a stepwise fashion from 55 °C to 70 °C in D1–D5 distillations of L-Bg-E to keep fluidity of the residue stream.

### 2.4. The Effect of EVT on Distillate Enrichment with Key Lavender Fragrance Molecules

The lavender extracts were submitted to MD experiments, which were performed with evaporator temperatures (EVT) ranging from 55 °C to 95 °C (five experiments per extract), at a constant pressure (1 Pa) and under a constant feed flow rate (FF) of 0.833 mL/min (Table 4). The results were analyzed in terms of contents of targeted oxygenates (mg/g) in the obtained distillates and the ratio of distillate stream mass to residue stream mass (D/R) obtained at every single step of the MD processes (D1–D5).

According to Tovar et al. [20], two parameters were important for effective separation of citral from lemongrass essential oil, i.e., EVT and FF. The last parameter, FF, was kept the same during both MD processes, while EVT was increased stepwise from 55 to 95 °C (Table 3). However, processing of L-Bg-E appeared to be more difficult in terms of a mass and heat transfer and required adjusting more parameters at the same time, compared to MD of L-Pl-E (Table 3). Different physicochemical properties of L-Bg-E, and its “congelation” over the distillation time, required maintaining a higher wiper basket speed of 350 rpm. Constant mixing of the falling film caused by the roller wiper action provided the heat transfer into deeper layers of the extracts more abundant with volatile molecules, and favored their concentrations in the vapor phase. The faster wiper basket movement (more rotations per minute) and increased evaporation efficiency resulted in intensified condensation. Hence, in order to ensure concentrations of the desired lavender fragrance components in the distillate stream, and to minimize material loss in the vapor phase and condense in the cold trap, the CT temperature was adjusted (changed from −5 °C to 6 °C in distillations D1–D5) at each stage of the L-Bg-E processing (Table 3). Low amounts of material condensed in the trap (lower than 8%) of the vacuum system previously found an indicator of the effective separation [20]. The yields of water and light oil (%wt_CT_), collected after each stage of MD of both lavender extracts did not exceed, in total, 8% (Table 4). However, the loss of light oil increased with the increase of the EVT, to the maximum level at 95 °C.

Additionally, with the highest applied EVT favoring highest recovery of the lower boiling fraction, the viscosity of a residual part increased and, consequently, the residue discharge temperature was increased up to 70 °C (Table 3). Further processing of L-Bg-E under selected conditions with higher EVT exceeding 95 °C can be unworkable because of the residue viscosity.

Results show that, with increasing EVTs, the percentage weight of the distillate stream (%wt_D_) also increased (Table 4). The highest values of %wt_D_ and, thus, the highest split ratio (D/R), were obtained for experiments D4 and D5, with EVTs of 85 °C and 95 °C, respectively. The highest was also the percentage weight of the cold trap fraction (%wt_CT_), referring to the content of water and light oil collected in the direct cooled cold trap (−80 °C). However, removal of the light oil fraction from L-Pl-E proceeding with increasing EVTs (55–95 °C) did not affect the viscosity of residuals as much as in the case of L-Bg-E. Hence, the RdT was quite low and remained unchanged over processing of L-Pl-E up to an EVT of 95 °C. A further slight increase in EVT might cause a further increase in %wt_D_ and enhance recovery of oxygenated compounds in the stream of distillates. In the example of citral, Tovar et al. [20] confirmed that with the highest applied EVT (60–120 °C) and the highest feed flow rate (1.5–4.5 mL/min), the concentration of this compound in the distillate stream doubled to 40.963 mg/mL compared to the initial concentration, proving the high product quality. However, according to Li et al. [27], the EVT exceeding 120 °C was found deleterious for scCO_2_ extract of *Artemisia annua* separated with MD under a similar vacuum (1.67 Pa). The combination of SFE and MD purification methods below 120 °C allowed producing high-quality essential oils, mainly composed of limonene, (*1S,5S*)-α-pinene, β-pinene, β-farnesene, α-caryophyllene, and γ-elemene, exhibiting antimicrobial and antioxidant activities [27].

The contents of target oxygenates quantified in lavender distillates obtained in the applied EVT range (55–95 °C) are listed in Table 5. The increase of EVT was crucial for the increase in contents of oxygenated monoterpenes and caryophyllene oxide in the distillate streams of both processed lavender extracts. Once the EVT increased up to 85 °C, the contents of 1,8-cineole, linalool, linalyl acetate, terpinen-4-ol, lavandulyl acetate, lavandulol, and caryophyllene oxide increased 2.0–2.4 times (L-Pl-D4) and 2.0–2.2 times (L-Bg-D4) in relation to the crude extracts (Table 5).

Linalool, lavandulyl acetate, and linalyl acetate, the most abundant oxygenated ingredients of L-Bg-E and L-Pl-E, differed in contents between both feedstocks. They were quantified at almost twice lower levels in L-Bg-E than in L-Pl-E (Table 5). The same applied to terpinen-4-ol, lavandulol, and caryophyllene oxide. In the case of L-Pl-E, the content of these three oxygenated compounds quantified by GC–FID were 59.16, 43.61, and 89.53 mg/g, respectively. The contents of linalool, lavandulyl acetate, and linalyl acetate gradually increased in the distillates across stages with a maximum in L-Pl-D4 (131.79, 92.53 and 185.73 mg/g, respectively) (Table 5). However, the increase of EVT from 85 °C to 95 °C caused a slight decrease in contents of those monoterpenoids (except from coumarins and caryophyllene oxide) compared to distillates L-Pl D4. The contents of less abundant monoterpenoids in L-Pl-E: 1,8-cineole (1.65 mg/g), terpinen-4-ol (8.96 mg/g), lavandulol (14.40 mg/g), and caryophyllene oxide (15.88 mg/g) also increased twice after MD at an EVT of 85 °C. A further increase in EVT by 10 °C caused a slight increase in those contents.

The content of 1,8-cineole was similar in both lavender extracts and was concentrated to the same extent (2.3-times for L-Bg-E and two-times in L-Pl-E) upon increasing the EVT to 85 °C. According to Table 5, the molecular distillation of L-Bg-E under the highest applied EVT of 95 °C generally caused a greater decrease in contents of monoterpenoids and caryophyllene oxide in the distillate compared to L-Pl-E. This might be an effect of dilution since the second heating DSC graph of the L-Bg-D5 distillate revealed a solid–liquid melting transition at 115 °C (Figure 5). The co-distilled higher-boiling components caused a slight cloudiness of L-Bg-D5 chilled to −4 °C.

The molecular distillation of L-Pl-E under the highest applied EVT temperature of 95 °C neither caused a significant change in D/R nor in the contents of the oxygenated monoterpenes compared to the process performed at 85 °C (Table 5). At the same time, it was noticed that the amount of the cold trap fractions (%wt_CT_) collected in a glass cylinder covering a steely cool finger (−80 °C) increased remarkably after the distillation process at 95 °C (experiment D5) of L-Pl-E, yielding, in total, 7.95%. The essential oil condensed with water and then separated as a lighter layer was the predominating cold trap material (4.53%). Thus, the use of EVT = 95 °C only slightly influenced the fraction L-Pl-D5, compared with L-Pl-D4. At the same time, lighter fragrance components were intensified in the vapor phase as the loss and were collected in the cold trap.

Coumarin and herniarin were the less volatile components amongst the analyzed oxygenated molecules (Table 3) co-distilled in some parts with oxygenated monoterpenes. According to the SFC analysis, it was confirmed that the yield of coumarin was higher than herniarin in both studied extracts (L-Pl-E and L-Bg-E). However, similar to the content of monoterpenoids, the content of both coumarins was almost twice higher in L-Pl-E compared to L-Bg-E (25.96 vs. 12.30 mg/g). Although both components were detected in distillates D1–D5 obtained from both feedstocks, their contents under processing conditions were lower than in the initial extract samples.

The other co-distilled components affecting the quality of oxygenated monoterpenes were pigments, which influenced the yellowish color of the distillates. Figure 6 depicts distillates (L-Pl-D4 and L-Bg-D4) obtained at EVT of 85 °C, light oil fractions collected after the process (L-Pl-O and L-Bg-O), and yellow–green residue fractions (L-Pl-R4 and L-Bg-R4, EVT = 85 °C) obtained in the process based on parameters listed in Table 4. The distillates and colorless light oil fractions were used in the following tests of quality and antimicrobial activities.

### 2.5. Quality Evaluation of Distillates and Light Oil Fractions

A non-targeted fingerprint HATR-FTIR analysis was performed to evaluate aromatic lavender concentrated fractions. Recently, vibrational spectroscopy methods, including mid- (MIR) and near-infrared (NIR), combined with chemometric data analyses, were used to confirm the identity and quality of lavender essential oils for commercial purposes [56,57]. Mid-infrared spectroscopy, primarily used qualitatively, provides structural characterization according to the functional group vibration and fingerprint region, which are crucial for molecular identification. The GC–MS is another technique, an alternative one, used for the quality assessment of concentrated fractions, i.e., distillates and light oils (cold trap fractions). The HATR-FTIR mid-infrared spectra of lavender distillates (L-Pl-D4 and L-Bg-D4) and light oil fractions (L-Pl-O and L-Bg-O) obtained from studied lavender extracts; L-Pl-E and L-Bg-E are depicted in Figure 7 and Figure 8, respectively.

The application of HATR-FTIR enabled the analysis of selected fractions in the form of ultra-thin films with no additional pre-treatment. HATR-FTIR absorption spectra showed characteristic bands identified previously in lavender essential oils [58,59]. Since major components of lavender oils are linalyl acetate and linalool, FTIR spectra are dominated by vibrational modes from those monoterpenoids. The carbonyl groups (C=O) present in linalyl acetate and lavandulyl acetate were characterized by peaks at ca. 1735 cm^−1^. The corresponding band in L-Pl-D4 had a small shoulder with a maximum at 1680 cm^−1^, indicating the formation of a hydrogen bond between C=O and –OH groups [57]. The area between 1100 and 1300 cm^−1^ included absorptions, representative for C-O stretching vibrations; those were documented at around 1239 cm^−1^, 1171 cm^−1^, 1112 cm^−1^, and 1110 cm^−1^, with some shifts in the 1176–1109 cm^−1^ region (Figure 7 and Figure 8). The other characteristic vibrational frequencies for linalool and linalyl acetate were associated with the vinyl group vibration (−C=CH_2_); however, the intensity of a related peak around 1646 cm^−1^ is very weak. Additionally, the −C=CH_2_ in-plane deformation vibration can be found at 1416 cm^−1^ as a weak band (Figure 7 and Figure 8). The feature characteristic of the O-H stretching vibration of alcohol functional groups present in linalool, its derivatives: 8-hydroxylinalool, furanoid linalool oxides, 3,7-dimethyl-1,5-octadiene-3,7-diol (terpendiol I), and lavandulol, could be found as a broadband, in the region of 3400–3500 cm^−1^. Those bands were found to be some of the strongest affecting the principal components in the *Lamiaceae* family group essential oils on the basis of ATR-FTIR and PCA analysis [57]. In the spectra of lavender distillates, L-Pl-D4 and L-Bg-D4 (Figure 7 and Figure 8), the signal of the O-H stretching mode was broad and more distinct compared to the spectra of corresponding light oils, which indicates hydrogen-bonding interactions between molecules. The differences in band positions, shapes, and intensities were also found in the area between 920 and 1240 cm^−1^ when comparing the spectra of lavender distillates and corresponding light oil fractions. The bonds, which have absorptions in the mentioned part of the fingerprint region, are those assigned to the C-O stretching vibrations, O-C-O from primary alcohols (i.e., lavandulol) and =C-H below 1000 cm^−1^. The bands at approximately 2800–3200 cm^−1^ may be related to C-H stretching and C=C-C ring vibrations, both documented as absorbing around 2950 cm^−1^ [56]. Since the last mentioned vibrations are attributed to molecular fragments of linalyl acetate, linalool, and its derivatives (Figure 3), which are major constituents of the obtained lavender fractions, they would not impart significant changes in the FTIR spectra.

The GC–MS semi-quantitative characterization results of distillates L-Pl-D4 and L-Bg-D4, and the corresponding light oil fractions L-PL-O and L-Bg-O, which were collected in the cold trap, are reported in Table 6. These multicomponent mixtures are characterized by a few major compounds in higher contents (20–70%). The major monoterpenoids of lavender fractions were linalool and linalyl acetate; however, there was a significant difference between the contents of each component when comparing fractions obtained from molecular distillations of L-Pl-E and L-Bg-E. The lowest linalool percentages were confirmed in L-Bg-D4 (7.58%) and L-Bg-O (17.44%), and, at the same time, the furanoid linalool oxides appeared in the highest content. The other late-eluting linalool derivatives/bioconversion products: 2,6-dimethyl-3,7-octadiene-2,6-diol, 2,6-dimethyl-1,7-octadiene-3,6-diol, 8-hydroxylinalool, 6,7-epoxylinalool, 3,7-dimethyl-1,7-octadiene-3,6-diol (terpendiol II) were found in higher amounts in L-Bg-D4 than in L-Pl-D4 (Figure 3, Table 6). Linalyl acetate was the most abundant component of the studied extracts, distillates, and light oil fractions; however, its percentage in distillate L-Bg-D4 was notably lower compared to L-Pl-D4 (Table 6). Since, in both distillates, physical refining caused a removal of heavy compounds, i.e., fatty acids and waxes, which diluted essential oil components in feedstocks, the late-eluting components remaining in L-Bg-D4 and L-Pl-D4 were coumarins, sesquiterpenes, and their derivatives (Figure 9). The identification of individual components using GC–MS and library spectra might not be reliable, particularly in the sesquiterpene region of the chromatogram where there were eluted, structurally-related components [60]. Some of the components eluted between 30 and 55 min (Figure 9) remained uncharacterized (Table 6). Nevertheless, according to a quantitative analysis by GC–FID, employing an external standard technique, caryophyllene oxide was quantified in both distillates, L-Pl-D4 and L-Bg-D4, at different levels, 34.75 and 13.80 mg/g (Table 6), respectively. Caryophyllene (and its oxidation product) was one of the few sesquiterpenes assigned in the light oil fractions (Table 6) using GC–MS.

### 2.6. Antimicrobial Activity

This is the first report to study MD fractions in terms of antimicrobial activities. As presented in Table 7, the distillates (L-Pl-D4, L-Bg-D4) and the related cold trap light oil fractions (L-Pl-O and L-Bg-O) isolated from two lavender scCO_2_ extracts showed antibacterial activity. Gram-positive bacteria were more susceptible to both distillates and light oils according to the minimum inhibitory concentration (MIC*)* (MIC = 0.5–4 mg/mL) when compared with *E. coli*, being Gram-negative bacterium (MIC = 8 mg/mL). The yeast strains were more susceptible to the distillates and light oils (MIC = 0.5–1 mg/mL) than the most bacterial strains. There were some differences between the antimicrobial activities of distillates (L-Pl-D4 and L-Bg-D4) and the corresponding light oil fractions. Predoi et al. [61] reported MIC and MBC values of <0.1 for essential oil hydrodistilled from *L. angustifolia*. However, the studied *L. angustifolia* hydro-distillated from southern Romania was characterized by a higher content of linalool (47.75%) when compared with L-Pl-D4 (19.79%) and L-Bg-D4 (7.58%).

Subsequently, the activity of the scCO_2_ crude extracts from lavender (L-Pl-E and L-Bg-E) against bacteria and yeasts was also examined. Gram-positive bacteria were found to be more-or-less susceptible to both extracts as compared to the distillates and light oils, depending on the bacterial species. The highest activity of both extracts was observed against *Bacillus subtilis* ATCC 6633 with MIC = 0.25 mg/mL (L-Pl-E) and MIC = 0.06 mg/mL (L-Bg-E). *Candida* spp. showed lower susceptibility to both extracts (MIC = 2–4 mg/mL) than to distillates and light oils.

It was found that all studied fractions (distillates and light oils) possessed bactericidal and fungicidal effects, confirmed by MBC minimum bactericidal concentration/MIC = 1–4 and minimum fungicidal concentration (MFC)/MIC = 1–2. It is generally accepted that antimicrobials are usually regarded as bactericidal or fungicidal if the MBC/MIC or MFC/MIC ratio is ≤4 [62]. However, it should be noted that the crude lavender extracts had bactericidal (MBC/MIC = 1–4) or bacteriostatic effects (MBC/MIC = 4–16), depending on the bacteria species, while both extracts exerted fungicidal effects (MFC/MIC = 1–2).

The reference substances, such as coumarin, herniarin (7-methoxycoumarin), linalool, linalyl acetate, caryophyllene oxide, lavandulol, and lavandulyl acetate were also used for antibacterial and antifungal activity tests (Table 8). Generally, lavandulol and lavandulyl acetate were mostly active against Gram-positive bacteria, with yeasts MIC ≤ 0.5 mg/mL showing the widest spectrum of activity. *B. subtilis* ATCC 6633 and *M. luteus* ATCC 10240 were found susceptible to all compounds included. They had bactericidal (MBC/MIC = 1–4) or bacteriostatic effects (MBC/MIC = 8–128), depending on the bacteria species, while all exerted fungicidal effects (MFC/MIC = 1–4).

## 3. Discussion

*Lavandula angustifolia* Mill. is a known source of essential oil (*Oleum Lavandulae*) with well-defined biological, antimicrobial, and therapeutic properties. Hence, there is a strong demand for high quality lavender products, desirable not only by the fragrance and perfume industries, but also in skincare and beauty products, pharmaceutical products, and in other forms of integrative medicine. According to Moussi Imane et al. [63] and Ciocarlan et al. [64], *L. angustifolia* essential oil demonstrates good activity against several bacterial species.

The essential oil of *L. angustifolia*, which is commonly produced through steam distillation or hydrodistillation as its variant, is one of the most extensively studied essential oils in terms of its antimicrobial activities [4,6,8,38,59,65]. Linalool and linalyl acetate are predominant lavender essential oil oxygenated monoterpenes (about 70% of the total composition) and their biological activities (e.g., anti-inflammatory [66], antibacterial [67,68], and antifungal [69]) have been evaluated individually. However, specific biological activities of the oil depend on the contributions of individual metabolites, classified as major and minor. Many scientific studies have compared different *L. angustifolia* essential oils, showing how strongly a geographic region of origin can influence the chemical composition of metabolites and, thus, contribute particular antimicrobial activities [65].

This study aimed to produce lavender fractions, rich in oxygenated monoterpenes from *L. angustifolia* supercritical extracts, using molecular distillation. Both SFE and MD techniques are gentle, free of toxic solvents, and could be used to design industrial processes. Recent applications of SFE coupled with MD have been developed mostly for enrichment of artemisia and turmeric scCO_2_ extracts in bioactive constituents of corresponding essential oils and further refinement of obtained extracts [15,27,28]. Although the SFE process of *L. angustifolia* performed under optimized conditions [13,14] was found to be the most advantageous technology, in terms of high total extraction yield and high recovery of precious oxygenated monoterpenes, there is no available procedure that combines SFE and MD, in regard to separating crucial lavender essential oil components from scCO_2_ extracts. Regarding the composition of lavender scCO_2_ extracts obtained from high density scCO_2_, they were characterized as mostly similar (closest) to starting the composition of the plant matter, with coumarins and fragrance molecules “fixed” with wax structures [13,70].

In this research, two molecular distillations were performed for scCO_2_ extracts obtained from *L. angustifolia* cultivated in two European countries: Poland and Bulgaria. The variations of the extracted plant material, owing to the geographic source, made this study more challenging, in terms of MD parameters that had to be adjusted individually to each of the studied lavender scCO_2_ extracts. The differences between yellow–green extracts, L-Bg-E and L-Pl-E, were reflected mostly in the compositions of the waxy structures, which affected their melting properties and fluidity. The different physicochemical properties of L-Bg-E and its “congelation” tendency over distillation times required maintaining higher wiper basket speeds, higher temperatures of the feed tanks and residue discharge, compared to processing parameters during distillation of L-Pl-E. However, both purification procedures were evaluated through five experiments according to the influence of the evaporator temperature increasing stepwise from 55 °C to 95 °C, with an interval of 10 °C. Monoterpenoids, coumarins, and caryophyllene oxide contents in distillates (D1–D5) and corresponding feedstreams (L-Bg-E and L-Pl-E) were analyzed quantitatively. The amount of material (light oil and water) collected in the cold trap glass cover (vacuum pump system guard) was also controlled after each experiment. The best results for high-quality distillates and process conditions ensuring efficient mass and heat transfer during molecular distillation of both lavender extracts were obtained at an EVT of 85 °C. In both distillates, L-Bg-D4 and L-Pl-D4, it was confirmed that the contents of the selected monoterpenoids, including crucial lavender fragrance ingredients, i.e., linalyl acetate, linalool, and lavandulyl acetate increased twice compared to corresponding starting crude extracts. This is due to removal of higher boiling waxy components. Moreover, distillates contain coumarins and their contents are at similar levels in L-Bg-D4 and L-Pl-D4, although, in the corresponding extracts, their amounts are higher. The light oil fractions (L-Bg-O and L-Pl-O) are abundant with target monoterpenoids, linalool derivatives, and some sesquiterpenes, while caryophyllene oxide is the highest boiling component of the fraction. Vibrational modes from major monoterpenoids dominated FTIR spectra, which additionally confirmed good quality of the distillates.

Selected distillates (L-Bg-D4 and L-Pl-D4), as well as standard reference materials (separate compounds), were also evaluated for their antimicrobial properties by determining the minimum inhibitory concentration (MIC) by the broth microdilution method. The antibacterial and antifungal activities of *L. angustifolia* Mill. can be explained by the presence of such components as linalool, linalyl acetate, lavandulol, geraniol, and eucalyptol [8]. In the present study, lavandulol and lavandulyl acetate were found to be active mostly against Gram-positive bacteria and yeasts with MIC ≤ 0.5 mg/mL. These compounds showed the widest spectrums of activity among the components included in the present study, also identified elsewhere in *L. angustifolia* Mill. [37]. All included reference substances had bactericidal (MBC/MIC = 1–4) and fungicidal effects (MFC/MIC = 1–4).

The light oils, similar to distillates, collected as cold trap fractions (L-Pl-O and L-Bg-O) during molecular distillations at 85 °C, were also studied for their antimicrobial properties. The light oil fractions (L-Bg-O and L-Pl-O) are abundant with target monoterpenoids, linalool derivatives, and some sesquiterpenes, while caryophyllene oxide is the highest boiling component of the fraction. Regarding antimicrobial activity of the distillates (L-Pl-D4, L-Bg-D4) and their related light oils (L-Pl-O, L-Bg-O), it should be noted that, generally, distillates proved to be somewhat more active than light oils, most probably due to their multicomponent nature (sesquiterpenoids and coumarins). However, the higher amount of linalool and linalyl acetate was still responsible for inhibitory effects against several bacterial species [71]. Distillates and oils exerted biocidal effects against bacteria and fungi (yeasts), and even higher susceptibility of *Candida* spp. to both distillates and light oils than to the crude extracts observed. The extracts revealed the highest activity against *B. subtilis* ATCC 6633 compared to other tested reference strains. L-Bg-E was more active (MIC = 0.06 mg/mL) compared to L-Pl-E (MIC = 0.25 mg/mL).

The presented data indicate the scCO_2_ extracts from *L. angustifolia* cultivated in Poland and Bulgaria as well as the corresponding fractions (distillates, light oils) may be regarded as a source of the valuable components with antimicrobial activity.

It was reported that biological activity of *Lavandula* species, including *L. angustifolia* Mill., depends on several factors, including the extraction procedure [72,73]. Recently, Garzoli et al. [74] studied the activity of essential oils and hydrolates from *L. angustifolia* Mill. grown in Italy against both Gram-positive (*Bacillus cereus*) and Gram-negative bacteria (*Escherichia coli*, *Acinetobacter bohemicus*, *Pseudomonas fluorescens*). Essential oils were found to possess high antibacterial activities with MICs ranging from 0.19 to 1.56% (*v*/*v*), in comparison, hydrolates did not show any inhibition of the bacterial growth at the tested concentrations due to the presence of only some volatile oil compounds.

## 4. Materials and Methods

### 4.1. Chemical and Reagents

Food grade CO_2_ ≥99.90% (Zakłady Azotowe Puławy S.A., Puławy, Poland) was applied to SFE. All solvents (dichloromethane and methanol) were of analytical grade and were purchased from J.T. Baker (Center Valley, PA, USA). The certified reference material, C7–C40 saturated n-alkanes (1000 μg/mL each component in hexane) were obtained from Supelco (Poznan, Poland). Linalyl acetate (≥97.0%), linalool (≥99.0%), caryophyllene oxide (≥99.0%), α-terpineol (≥98.5%), eucalyptol (≥99.0%), terpinen-4-ol, (primary reference standard), lavandulol (analytical standard), lavandulol acetate (analytical standard), coumarin (primary analytical standard), 7-methoxycoumarin (primary reference standard) were supplied by Merck (Poland). Carnauba and beeswax were purchased from ECOSPA (Warsaw, Poland).

### 4.2. Plants Materials

Lavender (*Lavandula angustifolia* Mill.) cultivars of Polish and Bulgarian origin were used in this study. Dry lavender flowers (L-Pl) and (L-Bg) were delivered by GRUPA INCO Company (Góra Kalwaria, Poland). Both raw materials were ground using a Retsch mill (SM100), sieved to 1 mm and subjected to SFE.

### 4.3. Supercritical Fluid Extraction

A pilot scale supercritical fluid extraction (SFE) with carbon dioxide was performed on a 40 L volume extractor (ELAB, Puławy, Poland) at the temperature of 40 °C, under the pressure of 30 Mpa, and the extraction time was 2 h. The CO_2_ consumption was 32.86 kgCO_2_/kg batch. The SFE resulted in a higher extraction yield (7.05 wt%) for L-Pl scCO_2_ extract compared to L-Bg scCO_2_ extract (6.33 wt%).

### 4.4. Molecular Distillation

A laboratory short-path distillation (SPD) system KDL-5 (UIC GmbH, Alzenau, Germany) was used to perform the molecular distillation of lavender scCO_2_ extracts. The wiped film molecular distillation model KDL 5 is a variation of falling film molecular distillation with agitation. The basic design of the wiped film variation of the (short path) distillation unit was a vertical double-jacketed cylindrical evaporator (4.8 dm^2^) with a cooled internal condenser (6.5 dm^2^) concentrically situated inside the cylinder. The evaporator surface (4.8 dm^2^), residue, and distillate sections were individually heatable with heating oil. The feed rate was continuously adjustable between 0 and 1.8 L/h. Due to the compact design of the feed system (heatable feed gear pump system with drive motor, double jacketed product feed vessel, roller wiper system with drive motor, and continuously adjustable speed), the areas between feed vessel and evaporator were heatable. For the optimal product distribution, the feed product was pumped on top of the rotating wiper basket plate (continuously adjustable sped from 200 to 500 rpm) to be wiped with rollers in the form of thin film falling down the surface of the glass evaporator. The glass cold trap (−80 °C) with an immersion cooling finger made of stainless steel was a part of the vacuum configuration and was used to trap and retain volatile molecules from the process to avoid pump oil contamination. The vacuum system was a set consisting of a rotary vane pump and an oil diffusion pump, put in series. Distillations were carried out at pressures around 1 Pa and were measured electronically by the Pirani style vacuum sensor.

### 4.5. Chemical Analysis

#### 4.5.1. Gas Chromatography Equipped with Tandem Mass Spectrometry (GC–MS/MS)

The identification and determination of the extract components were performed with an Agilent (Santa Clara, CA, USA) 7000 QQQ system (7890 gas chromatograph combined with triple-quadrupole MS with an electron ionization (EI) source). The capillary column DB-EUPAH UI (60 m × 0.25 mm i.d. × 0.25 μm) with helium (99.9999%) as a carrier gas at a flow rate of 0.6 mL/min was applied. Data acquisition was performed in the scan mode. The oven temperature was set to 60 °C at the starting point, and then gradually increased from 60 °C to 310 °C at 3 °C/min with a final hold of 3 min. Mass spectra were taken at 70 eV and the mass range (*m*/*z*) was 35–650. Identification of the components was carried out using MassHunter software (C.01.03), NIST Mass Spectral Library, and MS literature data. Retention indices (RIs) of the compounds were determined, relative to the retention times of a standard solution of n-alkanes for GC (C8–C20 and C21–C40). The relative percentage of the individual components was calculated based on the GC peak areas without the use of correction factors.

#### 4.5.2. Gas Chromatography Equipped with Flame Ionization Detection (GC–FID)

The quantitative analysis of the main fragrance components enriched in scCO_2_ extracts and distillate fractions (1,8-cineole, linalool, linalyl acetate, terpinen-4-ol, lavandulyl acetate, lavandulol, caryophyllene oxide) was performed by a capillary GC using a dual-channel Agilent 7820A GC system coupled with a flame ionization detector (FID). The volatiles were analyzed using the HP-5 column (30 m × 0.25 mm × 0.20 μm). The FID detector temperature was set at 300 °C and injector temperature was set at 250 °C. The oven temperature was programmed as follows: initial temperature 50 °C, raised to 260 °C (8 °C/min). and held at 260 °C for 5 min. High-purity helium (xx%) was the carrier and made-up gas with the flow rate of 1.6 and 25 mL/min, respectively. The flow relation for the FID detector was 30.0 mL/min for hydrogen and 400.0 mL/min for air. The injection of all samples (1 µL) was performed in a split mode (100:1) with a purge time of 0.75 min. An HP ChemStation ver. C.01.03, (Agilent, Santa Clara, CA, USA) was used for an instrument control and data analysis. The total analysis time was 27.25 min.

Calibration curves were obtained by analyzing eight standard solutions, which resulted in satisfactory linearity with high values for correlation coefficient (R^2^). The limit of detection (LOD) and limit of quantification (LOQ) were calculated as 3 and 10 SD/*a*, respectively, where SD is the standard deviation of the response and *a* is the slope of the calibration curve. The LOD and LOQ were 0.040 and 0.147 mg/mL for 1,8-cineole, 0.045 and 0.059 and 0.197 mg/mL for linalool, 0.058 and 0.194 mg/mL for linalyl acetate, 0.018 and 0.059 mg/mL for terpinen-4-ol, 0.134 and 0.445 mg/mL for lavandulyl acetate, 0.027 and 0.089 mg/mL for lavandulol, and 0.03 and 0.101 mg/mL for caryophyllene oxide, respectively. The solutions were prepared in dichloromethane.

#### 4.5.3. Supercritical Fluid Chromatography (SFC)

The SFC was used for the determination of coumarin and herniarin directly in extracts and distillate fractions after dilution in dichloromethane and methanol (1:1 *v*/*v*) on an Ultra-performance Convergence Chromatography (UPC^2^, Waters) system, with the Acquity UPC^2^ Fluoro-Phenyl column (3.0 × 100 mm × 1.7 μm) at 35 °C. The back pressure regulator set at 12.6 MPa controlled the outlet column pressure. The isocratic elution with a binary system of CO_2_ and methanol (98:2, *v*/*v*) was used for compound separation. The analyses were performed with the CO_2_/MeOH flow rate of 2.0 mL/min, injection volume of 2 μL, and UV wavelength at 270 nm for coumarin and 310 nm for herniarin.

The quantifications of coumarin and herniarin in lavender extracts and distilled fractions were performed by an external calibration curve constructed with standard solutions (coumarin; 1.28 mg/mL, herniarin; 1.20 mg/mL) prepared in dichloromethane and methanol (1:1, *v*/*v*). Calibration curve points (from 0.032 mg/mL to 0.64 mg/mL and from 0.03 to 0.60 mg/mL) were injected in triplicate. The results showed excellent linearity (R = 0.999–1.000). The LOD values were 2.98 μg/mL for coumarin and 3.29 μg/mL for herniarin, whereas LOQ were 11.24 and 10.73 μg/mL, respectively.

### 4.6. Differential Scanning Calorimetry

The thermal characteristics of the extract samples were measured on a TA Instruments Calorimeter, model DSC Q20. An indium (melting point 56.6 °C) was used as a standard for the equipment calibration. The extract sample (1–3 mg) was weighed into a closed aluminum pan and analyzed under nitrogen (50 mL/min) using a three-stage heating and cooling profile. The first heating from 20 to 150 °C, to remove the sample thermal history, was followed by an isotherm (at 150 °C) for 1 min, and then cooling to −50 °C, followed by the second isotherm for 1 min and second heating from −50 °C to 150 °C (1 min isotherm at 150 °C). All applied heating and cooling rates were 10 °C/min. The melting point range was determined using the DSC curves of the second heating cycle.

### 4.7. HATR-FTIR

The Thermo Scientific Smart Multi-Bounce HATR (Horizontal Attenuated Total Reflectance), Nicolet iS50 spectrometer (Thermo Scientific, Waltham, MA, USA), was applied to obtain high quality spectra of molecular distillation fractions of lavender oils. OMNIC^®^ software was used to acquire MIR spectra in absorbance mode with a wave range of 650–4000 cm^−1^. For each sample, 32 scans per sample were collected using spectral resolution of 4 cm^−1^. The crystal material was zinc selenide (ZnSe), with a 45-degree incidence angle. After the crystal was cleaned and infrared background was collected, each oil sample was poured onto the crystal and pressed to a thin film to cover the whole surface of the crystal.

### 4.8. Antimicrobial Tests

The assay of antibacterial and antifungal activity of the distillates and light oil fractions separated from the scCO_2_ extracts obtained from *Lavandula angustifolia* was performed by the broth microdilution method, according to European Committee on Antimicrobial Susceptibility Testing (EUCAST) recommendations [75]. The following reference strains were used in the study: *Staphylococcus aureus* ATCC 25923, *Staphylococcus aureus* ATCC 29213, *Staphylococcus aureus* ATCC BAA1707, *Bacillus subtilis* ATCC 6633, *Bacillus cereus* ATCC 10876, *Micrococcus luteus* ATCC 10240, *Enterococcus faecalis* ATCC 29212 (representatives of Gram-positive bacteria), *Escherichia coli* ATCC 25922 (representative of Gram-negative bacteria), *Candida albicans* ATCC 10231, *Candida glabrata* ATCC 90030, and *Candida krusei* ATCC 14243 (representatives of fungi belonging to yeast). All of the used microbial strains were first subcultured on Mueller-Hinton Agar (MHA) for bacteria or MHA with 2% glucose for fungi and incubated at 35 °C for 24 h. Microbial colonies were collected and suspended in sterile physiological saline to obtain inoculum of 0.5 McFarland standard, corresponding to 1.5 × 10^8^ CFU/mL (colony forming units) for bacteria and 5 × 10^6^ CFU/mL for fungi. The distillates and light oils were dissolved in DMSO to obtain the final concentration of 100 mg/mL.

The two-fold dilutions of distillates and light oils and extracts in Mueller-Hinton Broth (MHB) for bacteria or in RPMI with 2% glucose for fungi were prepared in 96-well polystyrene plates to obtain final concentrations ranging from 0.004 to 16 mg/mL. Next, 2 µL of a particular bacterial or fungal inoculum was added to each well, containing 200 µL of the serial dilution of distillates and light oils in the appropriate culture medium. After incubation at 35 °C for 24 h, the MIC was assessed spectrophotometrically as the lowest concentrations of distillates and light oils showing complete bacterial or fungal growth inhibition. Appropriate DMSO, growth, and sterile controls were carried out. Coumarin, herniarin (7-methoxycoumarin), linalool, linalyl acetate, caryophyllene oxide, lavandulol, and lavandulyl acetate were included as the standard antimicrobial plant substances active against Gram-positive bacteria, Gram-negative bacteria, and yeasts. The MBC or MFC were determined by removing 20 μL of the bacterial or fungal culture used for MIC determinations from each well and spotting this onto appropriate agar medium. The plates were incubated at 35 °C for 24 h. The lowest distillates and light oil concentrations with no visible bacterial or fungal growth were assessed as MBC or MFC, respectively. To determine the MIC, the absorbance was measured in a spectrophotometer at a wavelength of 600 nm. The experiments were performed in triplicate. On the basis of each MIC, MBC, and MFC value, the most common representative value, i.e., mode, was presented.

## 5. Conclusions

The supercritical extracts from *Lavandula angustifolia* Mill. are a diverse source of phytochemicals with principal fragrance constituents. Regarding the composition of extracts obtained with high density scCO_2_, they mostly resembled the starting plant matter, with coumarins and fragrance molecules admixed/combined with hydrophobic components of cuticle layers covering all aerial organs of plants, i.e., flowers. The physicochemical characterizations of scCO_2_ extracts (L-Pl-E, L-Bg-E) obtained on a pilot scale from two sources of *L. angustifolia*, Poland and Bulgaria, revealed differences in cuticle layer wax compositions, which, in the case of L-Bg-E, might have included esterified phenolic fraction. This higher melting fraction most likely gave the film of the cuticle covering lavender grown in Bulgaria specific barrier properties against UV radiation and uncontrolled water loss. One of the differences between the extracts concerned the content of linalool and its derivatives. The isomers of linalool oxide (tetrahydrofuran) and terpendiol (polyhydroxylated monoterpenes), representing the group of linalool derivatives, were included in a higher content in L-Bg-E comparing to L-Pl-E. The highest activities of both extracts were observed against *Bacillus subtilis* ATCC 6633 with MIC = 0.25 mg/mL (L-Pl-E) and MIC = 0.06 mg/mL (L-Bg-E).

In this research, for the first time, two procedures of molecular distillations were developed to refine the feedstocks: supercritical extracts (L-Pl-E and L-Bg-E) from *L. angustifolia* cultivated in Poland and Bulgaria. The variations of the extracted plant material, owing to the geographic source, made this study more challenging in terms of MD parameters having to be adjusted individually to each of the studied lavender scCO_2_ extracts. Each extract, L-Pl-E and L-Bg-E, was processed through five experiments, according to the influence of evaporator temperature increased stepwise from 55 °C to 95 °C at intervals of 10 °C. The best results and high-quality distillates (L-Pl-D4 and L-Bg-D4) were obtained at 85 °C (EVT), and it was confirmed that the linalyl acetate content increased from 51.54 mg/g (L-Bg-E) and 89.53 mg/g (L-Pl-E) to 118.41 and 185.42 mg/g, respectively, reaching an increase in its content of 2.3 and 2.1 times in both distillate streams. Once the EVT increased up to 85 °C, the contents of 1,8-cineole, linalool, terpinen-4-ol, lavandulyl acetate, lavandulol, and caryophyllene oxide doubled in the distillates streams. Those monoterpenoids, with predominating linalyl acetate and linalool, were the main active ingredients, as was studied in the extracts and distilled fractions.

Compared to the extracts, all distilled fractions with removed waxes and fatty acids by molecular distillation became valuable potential sources of lavender with antimicrobial activities against all studied *Candida* species. In general, scCO_2_ extracts are said to possess synergistic effects due to the content of the different groups of bioactive compounds. However, our studies revealed higher activities analyzed for enriched fractions (after MD) in comparison with crude extracts (without fractionation) in terms of specific pathogens. Only *Bacillus subtilis* was more sensitive to both extracts, rather than to the distillates and light oils.

## Figures and Tables

**Figure 1 molecules-27-01470-f001:**
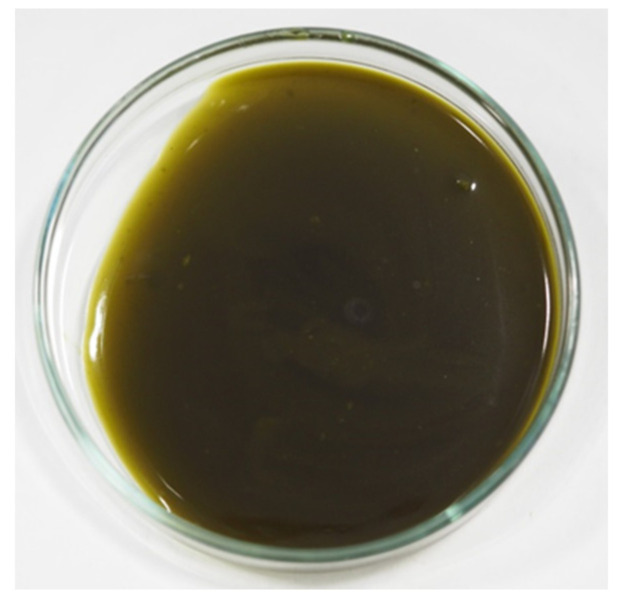
Lavender L-Pl-E scCO_2_ extract.

**Figure 2 molecules-27-01470-f002:**
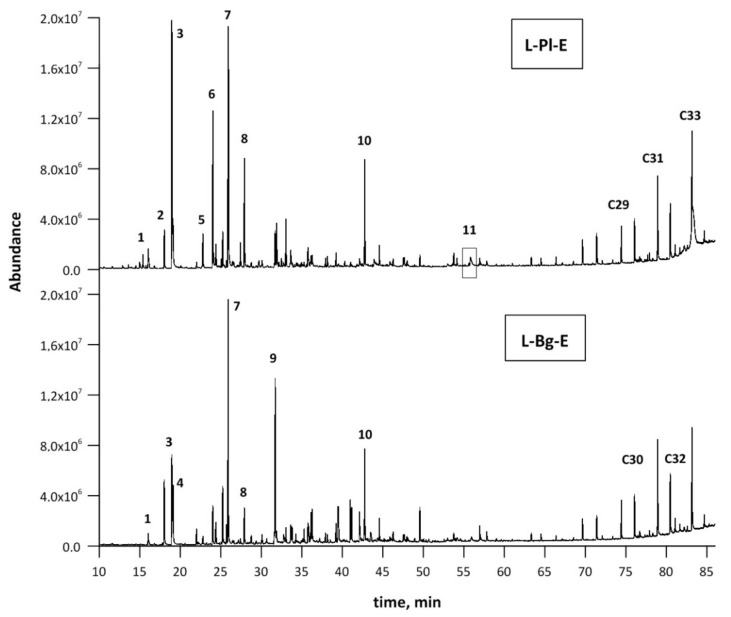
The GC–MS chromatograms of L-Pl-E and L-Bg-E samples. Peaks in order of elution: 3) eucalyptol, 4) *cis*-furan linalool oxide, 5) linalool, 6) *trans*-furan linalool oxide, 8) lavandulol, 9) terpinen-4-ol, 14) linalyl acetate, 16) lavandulyl acetate, 21) 3,7-dimethyl-1,7-octadiene-3,6-diol (terpendiol II), 40) caryophyllene oxide, 50) herniarin.

**Figure 3 molecules-27-01470-f003:**
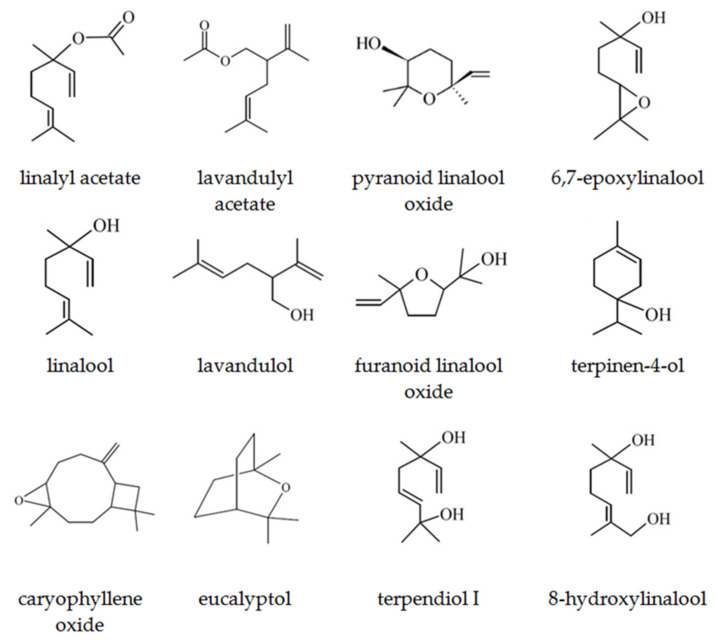
Chemical structures of the major and minor compounds identified in lavender scCO_2_ extracts.

**Figure 4 molecules-27-01470-f004:**
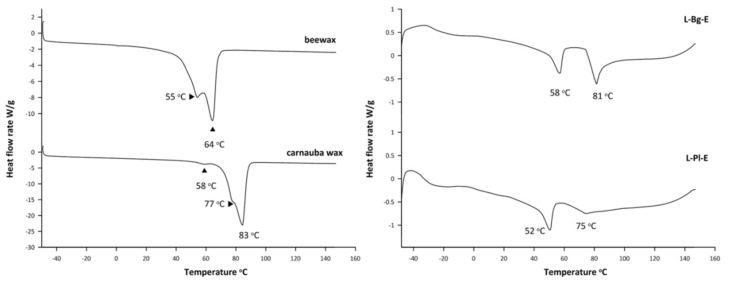
The second heating DSC graphs of lavender scCO_2_ extracts and commercial waxes: beewax and carnauba wax.

**Figure 5 molecules-27-01470-f005:**
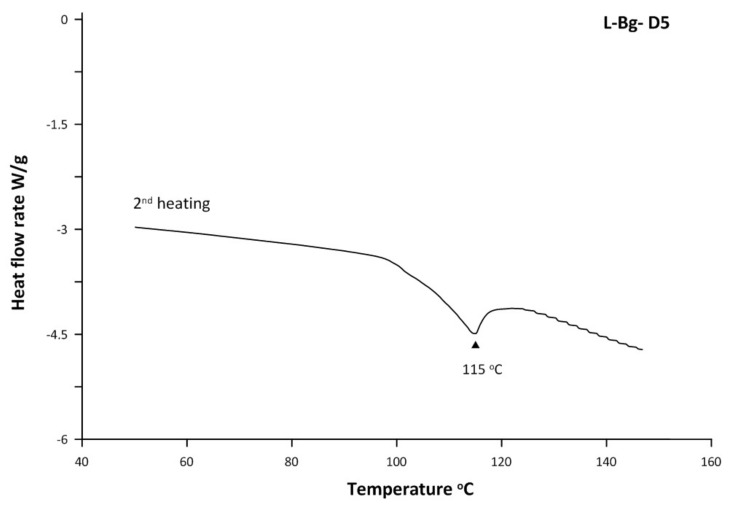
The second heating DSC graph of the L-Bg D5 distillate.

**Figure 6 molecules-27-01470-f006:**
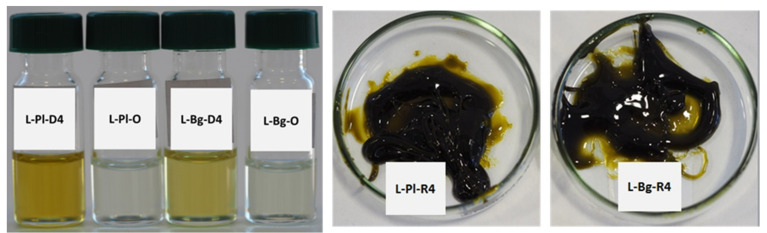
Fractions obtained from molecular distillation of L-Pl-E and L-Bg-E at EVT = 85 °C; O—light oil products; R—residues after molecular distillation.

**Figure 7 molecules-27-01470-f007:**
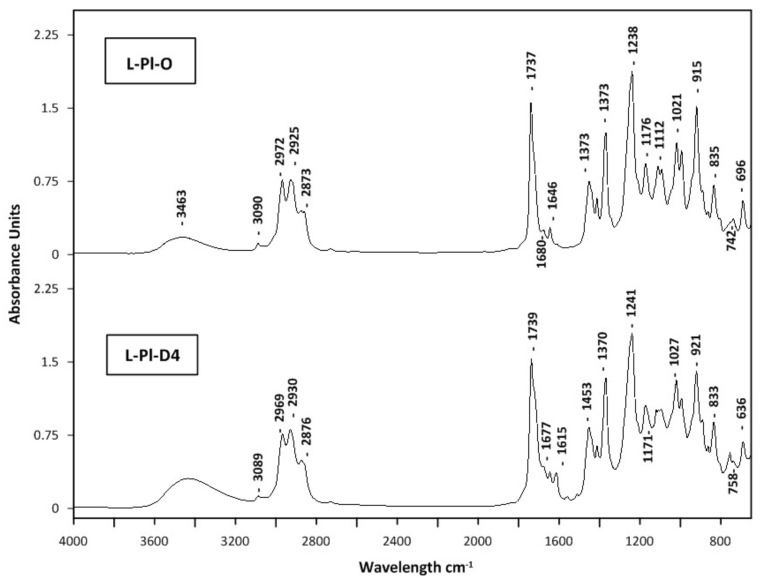
MIR spectra of lavender oil fractions; L-Pl-D4, L-Pl-O.

**Figure 8 molecules-27-01470-f008:**
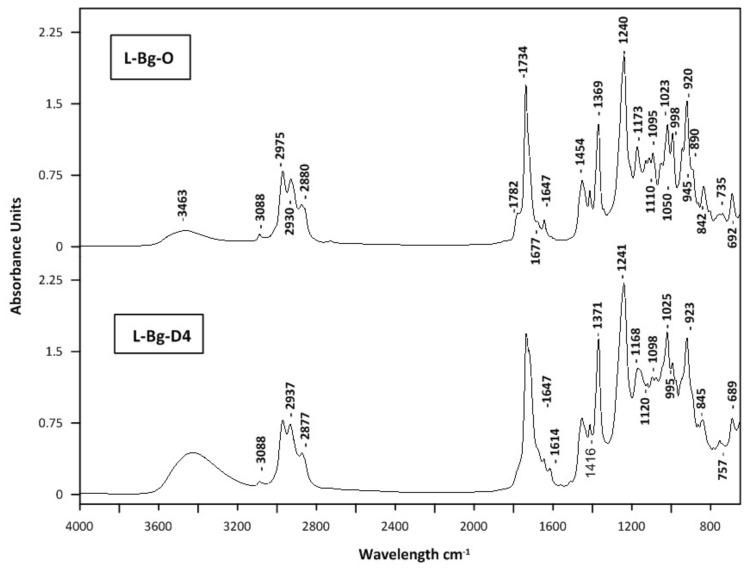
MIR spectra of lavender oil fractions; L-Bg-D4, L-Bg-O.

**Figure 9 molecules-27-01470-f009:**
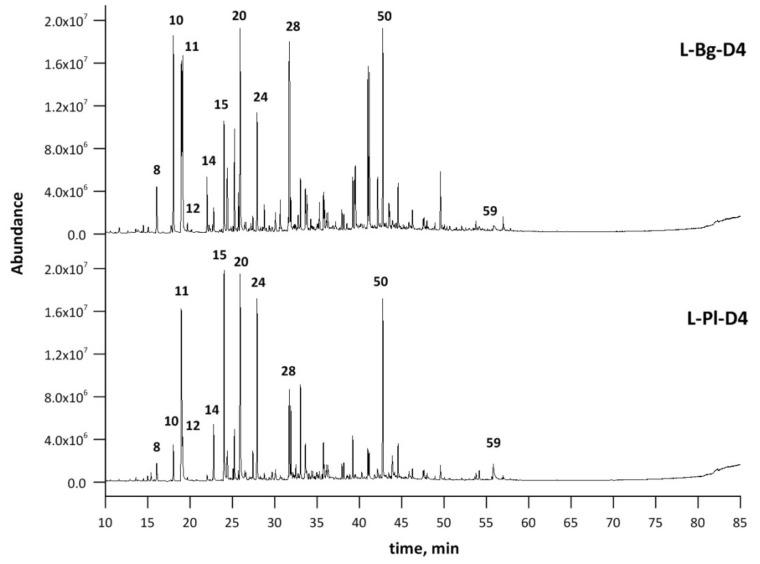
GC–MS chromatograms of L-Pl-D4 and L-Bg-D4 samples. Peaks in order of elution: 8) eucalyptol, 10) *cis*-furan linalool oxide, 11) linalool, 12) *trans*-furan linalool oxide, 14) lavandulol, 15) terpinen-4-ol, 20) linalyl acetate, 24) lavandulyl acetate, 28) 3,7-dimethyl-1,7-octadiene-3,6-diol (terpendiol II), 50) caryophyllene oxide, 59) herniarin.

**Table 1 molecules-27-01470-t001:** Chemical compositions of lavender scCO_2_ extracts.

No.	Compounds	RI	RT	Composition, %	
				L-Pl-E	L-Bg-E
1	(-)-Limonene	1114	14.99	0.26	-
2	*trans*-β-Ocimene	1123	15.39	0.55	-
3	Eucalyptol	1139	16.05	1.05	0.87
4	*cis*-Linalool oxide (furanoid)	1186	18.03	1.51	3.59
5	Linalool (3,7-dimethyl-1,6-octadien-3-ol)	1209	18.98	17.02	4.56
6	*trans*-Linalool oxide (furanoid)	1212	19.12	1.42	3.63
7	Lavender lactone	1278	22.01	0.26	0.79
8	Lavandulol	1295	22.79	1.40	0.43
9	Terpinen-4-ol	1323	24.02	6.02	2.15
10	Linalool oxide (pyranoid)	1332	24.40	0.95	1.55
11	α-Terpineol	1348	25.08	0.28	-
12	2,6-Dimethyl-3,7-octadiene-2,6-diol	1351	25.24	1.34	2.81
13	3,7-Dimethyl-1,5- octadiene-3,7-diol (terpendiol I)	1362	25.70	0.26	0.85
14	Linalool acetate	1367	25.93	25.68	12.78
15	Crypton	1402	27.42	0.87	0.28
16	(±)-Lavandulyl acetate	1413	27.91	3.96	1.86
17	Bornyl acetate	1433	28.76	-	0.40
18	*p*-Cumic aldehyde	1456	29.71	0.35	-
19	2,6-Dimethyl-1,7-octadiene-3,6-diol	1465	30.10	0.33	0.59
20	*p*-Cymen-7-ol	1501	31.59	-	0.54
21	3,7-Dimethyl-1,7-octadiene-3,6-diol, (terpendiol II)	1504	31.71	1.41	8.16
22	Longicyclene	1508	31.90	1.61	0.7
23	α-Bergamotene	1523	32.50	0.24	-
24	β-Caryophyllene	1537	33.04	1.76	0.67
25	β-Farnesene	1551	33.64	0.89	1.13
26	2,6-Dimethyl-2,7-octadiene-1,6-diol (8-hydroxylinalool)	1555	33.79	0.21	1.02
27	Nerolidol	1592	35.29	-	0.69
28	*cis*-2-Methyl-3-oxo-cyclohexanebutanal	1604	35.76	0.86	0.92
29	*cis*-9-Tetradecen-1-ol	1607	35.86	0.23	0.65
30	Epoxylinalool	1614	36.14	0.49	1.55
31	Farnesyl formate	1618	36.28	0.5	1.76
32	Nerolidyl acetate	1661	37.93	0.32	0.48
33	α-Santalol	1667	38.17	0.38	0.4
34	α-Bisabolol	1694	39.22	0.44	0.84
35	NI ^a^	1701	39.46	-	1.51
36	NI ^a^	1703	39.54	-	1.67
37	NI ^a^	1743	40.99	-	2.06
38	NI ^a^	1747	41.16	-	1.38
39	(1S,2S,4S)-Trihydroxy-*p*-menthane	1774	42.14	0.25	1.88
40	β-Caryophyllene oxide	1791	42.77	3.8	4.54
41	Coumarin	1821	43.92	0.18	-
42	*tau*-Cadinol	1842	44.58	0.66	1.01
43	Ledene oxide-(II)	1891	46.27	0.25	0.43
44	α-Santalol	1929	47.55	0.27	0.3
45	β-Santalol	1932	47.65	0.31	
46	Hexahydrofarnesyl acetone	1942	47.98	0.2	-
47	Isolongifolol	1990	49.58	0.39	1.55
48	Neryl acetate	2121	53.76	0.51	0.43
49	Gerany-p-cymene	2134	54.15	0.27	0.28
50	Herniarin	2189	55.83	1.13	0.38
51	Neoisolongifolene, 8-oxo	2228	56.98	0.21	0.7
52	7-Hydroxyfarnesene	2257	57.83	-	0.46
53	11-Methyltricosane	2453	63.33	0.27	0.31
54	Pentacosane, C25	2497	64.54	0.22	0.31
55	2-Methylhexacosane	2568	66.38	0.26	-
56	Heptacosane, C27	2697	69.66	0.81	0.95
57	1-Iodo-docosane	2769	71.40	1.43	1.21
58	Nonacosane, C29	2897	74.45	1.28	1.78
59	Tetracosane 1-iodo	2969	76.08	1.62	1.97
60	Triacontane, C30	2996	76.72	-	0.3
61	1-Heptacosanol	3051	77.92	0.3	0.29
62	Hentriacontane, C31	3097	78.93	3.14	4.68
63	Methyltriacontane	3169	80.48	2.23	3.22
64	Dotriacontane, C32	3197	81.08	0.46	0.73
65	Tritriacontane, C33	3297	83.16	4.57	5.43

Compounds identified by mass spectrum and retention index, area (%) calculated from the total ion current in GC/MS. RI: retention indices, calculated; ^a^ No reliable RI data. NI: not identified.

**Table 2 molecules-27-01470-t002:** Boiling and melting points of oxygenated monoterpenes, caryophyllene oxide, and coumarins.

Compounds	Formula	Molecular Weight, g/mol	Boiling Point, °C	Melting Point, °C
Coumarin	C_9_H_6_O_2_	146.14	301.7	68–70
1,8-Cineole	C_10_H_18_O	154.24	176.5	2.9
Linalool	C_10_H_18_O	154.25	198	<−20
Terpinen-4-ol	C_10_H_18_O	154.25	211–213	-
Lavandulol	C_10_H_18_O	154.26	229–230	-
Herniarin	C_10_H_8_O_3_	176.16	335.3	117–121
Linalyl acetate	C_12_H_20_O_2_	196.29	220	<25
Lavandulyl acetate	C_12_H_20_O_2_	196.29	228–229	−17.12
Caryophyllene oxide	C_15_H_24_O	220.35	279.7	60–62

**Table 3 molecules-27-01470-t003:** Conditions for molecular distillation of L-Bg-E and L-Pl-E.

Parameters							
Exp.	FT, °C	RdT, °C	EVT, °C	CT, °C	FF, mL/min	p, Pa	rpm
Bulgarian lavender							
L-Bg D1	50	55	55	−5	0.833	~1 Pa (10^−2^ mbar)	350
L-Bg D2		58	65	−5			
L-Bg D3		63	75	0			
L-Bg D4		68	85	6			
L-Bg D5		70	95	6			
Polish lavender							
L-Pl D1	45	55	55	10	0.833	~1 Pa (10^−2^ mbar)	200
L-Pl D2			65				
L-Pl D3			75				
L-Pl D4			85				
L-Pl D5			95				

FT—feed tank temperature; RdT—residue discharge temperature; EVT—evaporator temperature; CT—condenser temperature; FF—feed flow rate; p—pressure; D—distillate; rpm—wiper basket speed.

**Table 4 molecules-27-01470-t004:** Parameters to evaluate the molecular distillation of lavender scCO_2_ extracts.

EVT, °C	CT, °C	Fraction	%wt_D_	%wt_R_	D/R	%wt_CT_		%wt	Rec.
						Oil	Water		
Bulgarian lavender									
55	−5	L-Bg-D1	15.23	75.73	0.20	1.73	3.46	3.85	20.2
65	−5	L-Bg-D2	20.12	70.54	0.29	1.88	3.38	4.08	30.3
75	0	L-Bg-D3	34.87	55.51	0.63	2.32	3.54	3.76	64.0
85	6	L-Bg-D4	39.28	51.12	0.77	2.20	3.50	3.90	81.4
95	6	L-Bg-D5	44.65	44.86	1.00	3.12	3.25	4.12	82.8
Polish lavender									
55	10	L-Pl-D1	19.13	70.31	0.27	3.32	3.12	4.12	23.9
65		L-Pl-D2	24.86	64.43	0.39	3.15	3.31	4.25	37.2
75		L-Pl-D3	35.33	53.81	0.66	3.23	3.25	4.38	61.2
85		L-Pl-D4	44.44	44.06	1.01	3.70	3.30	4.50	86.6
95		L-Pl-D5	45.28	42.05	1.08	4.53	3.42	4.72	88.1

EVT—evaporator temperature; CT—condenser temperature; %wt—material loss, %wt_D_—the percentage weight of distillate stream; %wt_R_—the percentage weight of residue stream; D/R—ratio of distillate to residue; %wt_CT_—the percentage weight of cold trap fraction; Rec.—recovery.

**Table 5 molecules-27-01470-t005:** The fractionation results of L-Bg-E and L-PL-E during molecular distillation.

Stream	EVT, °C	Compounds								
		1 *	2 *	3 **	4 **	5 **	6 **	7 **	8 **	9 **
		Content, mg/g								
L-Bg-E	-	7.00	5.30	1.85	19.60	51.54	2.32	20.38	7.46	6.02
D1	55	8.48	3.79	3.03	25.10	75.80	3.22	25.85	9.53	8.22
D2	65	6.52	4.11	3.55	31.34	82.24	3.54	32.12	11.51	9.64
D3	75	5.46	4.25	4.02	37.52	107.45	4.23	36.76	14.38	11.61
D4	85	5.66	4.43	4.20	39.24	118.41	4.80	45.55	18.14	13.80
T-C-D4		0.80	0.80	2.30	2.00	2.30	2.10	2.20	2.40	2.30
D5	95	5.59	4.85	3.74	34.94	105.40	4.27	40.55	16.14	12.20
L-Pl-E	-	14.89	11.07	1.65	59.16	89.53	8.96	43.61	14.40	15.88
D1	55	14.11	7.58	2.48	75.34	127.54	10.32	50.45	15.81	19.65
D2	65	12.84	6.53	2.73	95.39	146.76	12.87	65.39	19.25	26.84
D3	75	10.49	5.12	3.07	112.80	165.87	15.89	88.22	24.32	29.32
D4	85	8.95	4.30	3.22	131.79	185.73	18.95	92.53	29.67	34.75
T-C-D4		0.60	0.40	2.00	2.20	2.10	2.10	2.10	2.20	2.10
D5	95	9.12	4.24	3.28	130.23	184.42	19.03	93.15	30.20	34.87

* Coumarins on the basis of UPC^2^ analysis (1—coumarin; 2—herniarin); ** oxygenated compounds based on GC–FID analysis (3—1,8-cineole; 4—linalool; 5—linalyl acetate; 6—terpinene-4-ol; 7—lavandulyl acetate; 8—lavandulol; 9—caryophyllene oxide), T-C-D4, number of times that components 1–10 were concentrated in the distillate (D4) related to their concentrations in the corresponding extract samples (L-Bg-E or L-Pl-E).

**Table 6 molecules-27-01470-t006:** The GC–MS chemical composition of distillates L-Pl-D4 and L-Bg-D4 and the corresponding light oils fractions, L-Pl-O and L-Bg-O.

No.	Compound	RI	RT	Composition, %			
				L-Pl-D4	L-Bg-D4	L-Pl-O	L-Bg-O
1	α-Pinene	1022	11.59	-	-	0.21	0.48
2	Camphene	1075	13.52	-	-	0.30	0.22
3	(-)-β-Pinene	1082	13.81	-	-	0.36	0.17
4	1-Octen-3-ol	1095	14.43	-	0.20	0.38	0.53
5	3-Octanone	1101	15.02	-	-	0.39	0.42
6	(-)-Limonene	1113	14.98	-	-	0.73	-
7	*trans*-β-Ocimene	1123	15.38	0.26	-	1.06	-
8	Eucalyptol	1139	16.05	0.84	1.80	3.27	4.23
9	Sabinene hydrate	1179	17.69	-	-	-	0.27
10	*cis*-Linalool oxide (furanoid)	1186	18.03	1.19	5.54	2.61	10.72
11	Linalool (3,7-dimethyl-1,6-octadien-3-ol)	1209	18.98	19.21	8.38	28.34	17.44
12	*trans*-Linalool oxide (furanoid)	1212	19.13	1.20	4.89	2.57	9.21
13	Lavender lactone	1278	22.01	-	1.32	0.41	2.49
14	Lavandulol	1295	22.78	1.84	0.71	1.52	0.87
15	Terpinen-4-ol	1323	24.01	7.17	3.06	7.56	4.20
16	Linalool oxide (pyranoid)	1332	24.40	1.05	2.33	1.42	2.87
17	α-Terpineol	1348	25.09	0.31	-	0.19	-
18	2,6-Dimethyl-3,7-octadiene-2,6-diol	1352	25.24	1.41	2.64	-	-
19	3,7-Dimethyl-1,5- octadiene-3,7-diol (terpendiol I)	1362	25.70	0.26	0.92	-	0.87
20	Linalool acetate	1368	25.94	30.51	20.84	36.42	32.34
21	NI	1380	26.46	0.31	-	-	0.21
22	Isobornyl formate	1392	26.52	-	-	0.29	-
23	Crypton	1402	27.42	0.80	-	1.07	0.55
24	(±)-Lavandulyl acetate	1413	27.91	5.17	3.02	4.50	3.99
25	Bornyl acetate	1433	28.76	-	0.66	0.22	1.03
26	*p*-Cumic aldehyde	1456	29.69	0.29	-	0.36	0.21
27	2,6-Dimethyl-1,7-octadiene-3,6-diol	1465	30.08	0.48	0.65	-	-
28	3,7-Dimethyl-1,7-octadien-3,6-diol, (terpendiol II)	1504	31.71	2.63	8.87	-	0.69
29	Longicyclene	1508	31.90	2.05	1.00	1.66	1.02
30	α-Bergamotene	1523	32.49	0.33	-	0.25	-
31	β-Caryophyllene	1537	33.04	2.68	1.33	2.73	1.69
32	β-Farnesene	1551	33.62	1.49	1.41	0.19	0.59
33	2,6-Dimethyl-2,7-octadiene-1,6-diol (8-hydroxylinalool)	1555	33.78	0.29	1.17	-	-
34	Limonene-1,2-diol	1569	34.36	0.22	-	-	-
35	Car-3-en-5-one	1573	34.49	0.25	-	-	-
36	Nerolidol	1592	35.28	0.25	0.60	-	-
37	*cis*-2-Methyl-3-oxo- cyclohexanebutanal	1604	35.75	1.18	0.99	-	-
38	*cis*-9-Tetradecen-1-ol	1607	35.86	0.39	0.81	-	-
39	Epoxylinalool	1614	36.13	0.35	-	-	-
40	Farnesyl formate	1618	36.28	0.55	0.50	-	-
41	Nerolidyl acetate	1661	37.93	0.43	0.54	-	-
42	α-Santalol	1667	38.17	0.49	-	-	-
43	α-Bisabolol	1694	39.21	1.18	1.24	-	-
44	NI	1705	39.46	-	1.46	-	-
45	NI	1716	39.54	-	1.51	-	-
46	Ascaridole	1723	40.28	0.24	0.24	-	-
47	NI	1743	41.00	0.95	4.11	-	-
48	NI	1747	41.15	0.54	3.75	-	-
49	(1S,2S,4S)-Trihydroxy-*p*-menthane	1774	42.13	0.45	1.53	-	-
50	β-Caryophyllene oxide	1791	42.77	5.63	5.30	0.98	1.27
51	NI	1811	43.47	-	0.82	-	-
52	Coumarin	1823	43.88	1.10	-	-	-
53	*tau*-Cadinol	1842	44.57	0.94	1.07	-	-
54	Longifolene aldehyde	1880	45.87	0.30	-	-	-
55	Ledene oxide-(II)	1891	46.27	0.31	0.60	-	-
56	Isolongifolol	1990	49.57	0.46	1.41	-	-
57	Neryl acetate	2121	53.76	0.25	0.24	-	-
58	Gerany-p-cymene	2134	54.16	0.29	-	-	-
59	Herniarin	2189	55.82	1.38	0.55	-	-

**Table 7 molecules-27-01470-t007:** Antimicrobial activity of distillates D4 and light oil fractions resulting from MD of lavender scCO_2_ extracts as well as crude extracts assessed as MIC, MBC, or MFC.

Microorganisms	L-Pl-D4		L-Pl-O		L-Bg-D4		L-Bg-O		L-Pl-E		L-Bg-E	
Gram-positive bacteria	MIC	MBC	MIC	MBC	MIC	MBC	MIC	MBC	MIC	MBC	MIC	MBC
*Staphylococcus aureus* ATCC 25923	1	2	2	4	2	2	4	8	2	4	2	4
*Staphylococcus aureus* ATCC 29213	2	4	4	8	4	4	4	8	4	8	8	8
*Staphylococcus aureus* ATCC BAA1707	2	4	4	8	4	8	4	8	2	8	4	8
*Bacillus subtilis* ATCC 6633	2	2	1	4	1	4	2	4	0.25	4	0.06	4
*Bacillus cereus* ATCC 10876	2	2	4	4	2	2	4	4	1	16	2	16
*Micrococcus luteus* ATCC 10240	0.5	2	1	4	1	2	1	4	0.5	4	0.5	4
*Micrococcus faecalis* ATCC 29212	1	4	1	4	2	8	2	8	1	16	4	16
Gram-negative bacteria	MIC	MBC	MIC	MBC	MIC	MBC	MIC	MBC	MIC	MBC	MIC	MBC
*Escherichia coli* ATCC 25922	8	8	8	8	8	16	8	16	8	16	16	16
Yeast strains	MIC	MFC	MIC	MFC	MIC	MFC	MIC	MFC	MIC	MFC	MIC	MFC
*Candida albicans* ATCC 10231	0.5	1	0.5	1	0.5	1	0.5	1	2	4	4	4
*Candida glabrata* ATCC 90030	1	2	1	2	1	2	1	2	2	4	4	8
*Candida krusei* ATCC 14243	1	1	1	1	1	2	1	2	2	4	4	8

MIC, MBC, and MFC were expressed as mg/mL. The representative data (mode) are presented.

**Table 8 molecules-27-01470-t008:** Antimicrobial activity of the selected compounds identified in *Lavandula angustifolia* Mill. assessed as MIC, MBC, or MFC.

Microorganisms	A		B		C		D		E		F		G	
Gram-positive bacteria	MIC	MBC	MIC	MBC	MIC	MBC	MIC	MBC	MIC	MBC	MIC	MBC	MIC	MBC
*Staphylococcus aureus* ATCC 25923	1	2	1	1	2	4	2	2	1	4	0.5	1	0.5	1
*Staphylococcus aureus* ATCC 29213	0.25	4	2	4	2	4	2	2	1	2	0.5	1	1	2
*Staphylococcus aureus* ATCC BAA1707	0.25	4	2	4	2	4	2	4	1	4	0.5	1	1	2
*Bacillus subtilis* ATCC 6633	1	4	0.125	2	0.5	2	2	2	0.5	4	0.06	0.5	0.06	2
*Bacillus cereus* ATCC 10876	1	4	0.25	4	2	4	2	4	1	4	0.5	4	1	4
*Micrococcus luteus* ATCC 10240	0.5	2	0.5	4	0.25	1	0.5	2	0.03	4	0.125	1	0.125	1
*Micrococcus faecalis* ATCC 29212	1	4	2	4	1	4	0.5	2	0.5	4	0.5	2	2	8
Gram-negative bacteria	MIC	MBC	MIC	MBC	MIC	MBC	MIC	MBC	MIC	MBC	MIC	MBC	MIC	MBC
*Escherichia coli* ATCC 25922	2	4	0.5	4	1	4	2	4	2	4	1	1	1	2
Yeast strains	MIC	MFC	MIC	MFC	MIC	MFC	MIC	MFC	MIC	MFC	MIC	MFC	MIC	MFC
*Candida albicans* ATCC 10231	1	2	0.5	1	1	1	0.5	1	0.5	2	0.5	0.5	0.5	1
*Candida glabrata* ATCC 90030	1	2	0.5	1	1	1	1	2	0.5	2	0.5	0.5	0.5	1
*Candida krusei* ATCC 14243	1	4	1	2	1	1	1	2	1	2	0.5	0.5	0.25	1

MIC, MBC, and MFC are expressed as mg/mL. The representative data (mode) are presented; A—coumarin; B—herniarin (7-methoxycoumarin); C—linalool; D—linalyl acetate; E—caryophyllene oxide; F—lavandulol; G—lavandulyl acetate.

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
