# Peer review of "Molecular Distillation of Lavender Supercritical Extracts: Physicochemical and Antimicrobial Characterization of Feedstocks and Assessment of Distillates Enriched with Oxygenated Fragrance Components"

_molecules, 2022, doi:10.3390/molecules27051470_

Round 1
Reviewer 1 Report
This manuscript entitled “Molecular Distillation of Lavender Supercritical Extracts: Physicochemical and Antimicrobial Characterization of Feedstocks and Assessment of Distillates Enriched with Oxygenated Fragrance Components” is well documented and has provided new insights into essential oil research. However, I suggest that authors shorten the write-up because the manuscript seems unnecessarily descriptive.
- Also, the authors are suggested lowering the similarity index, which is 19% now.
- Please check chemical names and focus on the stereochemistry of molecules correctly. For your reference, please use ChemSpider or SciFinder-CAS. For example, which isomers of terpinen-4-ol, and lavandulol, you have mentioned in Table 2.
- In Table 2, please correctly mention the chemical formula (values of C, H, O should be subscript.
- In line 624, please make bacterial strain Bacillus subtilis
- Did you examine the antimicrobial zone of inhibition (ZoI)? How are the results? Usually, MIC is determined after evaluating the ZoI.
- In Table 7, your MIC and MBC values are very high as compared to reported literature on similar work. In Table 8, you did not mention the concentration of MIC, and MBC. Please re-confirm your data and verify the literature.
Author Response
Dear Reviewer 1,
We would like to thank You for the review. Please find the changes in revised manuscript and answers in word doc.
Best regards,
Agnieszka Dębczak

Reviewer 2 Report
The manuscript "Molecular Distillation of Lavender Supercritical Extracts: Physicochemical and Antimicrobial Characterization of Feedstocks and Assessment of Distillates Enriched with Oxygenated Fragrance Components" presents the results of the characterization of lavender extracts. These are interesting and innovative studies, in particular the method of obtaining lavender extracts, which allows obtaining extracts with high biological activity.
Detailed comments:
line 31 - there should be bacteria
Please let me know why such types of microorganisms were selected for testing? It would be worth adding other G (-) bacteria and molds.
In subsequent studies, I suggest adding an analysis of antioxidant activity.
The authors did not use any statistical analysis of the data, in the case of the chemical composition analysis (Table 6), PCA analysis can be performed.
line 875-876 no indices at powers.
Why was MHB with 2% glucose used and not RPMI used in the fungal study?
line 883 - at what wavelength?
Author Response
Dear Reviewer 2,
We would like to thank You for the review. Please find the changes in revised manuscript and answers in word doc.
Best regards,
Agnieszka Dębczak

Reviewer 3 Report
The paper titled Molecular Distillation of Lavender Supercritical Extracts: Physicochemical and Antimicrobial Characterization of Feedstocks and Assessment of Distillates Enriched with Oxygenated Fragrance Components is interesting and it shows a lot of work of Authors. However, the content of the paper could be devided in two papers.
1) Could be good to add the photo of bilgarian lavender, as you showed polish lavender on Fig. 1.
2) Figure 3 - In my opinion the chemical structure of the minor compounds in lavender extracts is not necessary.
3)In my opinion, Fig. 5 could be deleted.
Author Response
Dear Reviewer 3,
We would like to thank You for the review. Please find the changes in revised manuscript and answers in word doc.
Best regards,
Agnieszka Dębczak

Round 2
Reviewer 2 Report
The corrections have been made in the text, I have no more comments.